# Influence of Extracellular Matrix Components on the Differentiation of Periodontal Ligament Stem Cells in Collagen I Hydrogel

**DOI:** 10.3390/cells12192335

**Published:** 2023-09-22

**Authors:** Alexey A. Ivanov, Alla V. Kuznetsova, Olga P. Popova, Tamara I. Danilova, Andrey V. Latyshev, Oleg O. Yanushevich

**Affiliations:** 1Laboratory of Molecular and Cellular Pathology, A.I. Evdokimov Moscow State University of Medicine and Dentistry, 20 Delegatskaya Str., 127473 Moscow, Russia; avkuzn@list.ru (A.V.K.); petrovnapopova@rambler.ru (O.P.P.); danilova.tam@yandex.ru (T.I.D.); doctor.latyshev@gmail.com (A.V.L.); 2Koltzov Institute of Developmental Biology, Russian Academy of Sciences, 26 Vavilov Str., 119334 Moscow, Russia; 3Department of Periodontology, A.I. Evdokimov Moscow State University of Medicine and Dentistry, 20 Delegatskaya Str., 127473 Moscow, Russia; olegyanushevich@me.com

**Keywords:** decellularized extracellular matrix, periodontal ligament stem cells, collagen I hydrogel, hyaluronic acid, fibronectin, laminin

## Abstract

Regeneration of periodontal tissues requires an integrated approach to the restoration of the periodontal ligament, cementum, and alveolar bone surrounding the teeth. Current strategies in endogenous regenerative dentistry widely use biomaterials, in particular the decellularized extracellular matrix (dECM), to facilitate the recruitment of populations of resident cells into damaged tissues and stimulate their proliferation and differentiation. The purpose of our study was to evaluate the effect of the exogenous components of the extracellular matrix (hyaluronic acid, laminin, fibronectin) on the differentiation of periodontal ligament stem cells (PDLSCs) cultured with dECM (combinations of decellularized tooth matrices and periodontal ligament) in a 3D collagen I hydrogel. The immunohistochemical expression of various markers in PDLSCs was assessed quantitatively and semi-quantitatively on paraffin sections. The results showed that PDLSCs cultured under these conditions for 14 days exhibited phenotypic characteristics consistent with osteoblast-like and odontoblast-like cells. This potential has been demonstrated by the expression of osteogenic differentiation markers (OC, OPN, ALP) and odontogenic markers (DSPP). This phenomenon corresponds to the in vivo state of the periodontal ligament, in which cells at the interface between bone and cementum tend to differentiate into osteoblasts or cementoblasts. The addition of fibronectin to the dECM most effectively induces the differentiation of PDLSCs into osteoblast-like and odontoblast-like cells under 3D culture conditions. Therefore, this bioengineered construct has a high potential for future use in periodontal tissue regeneration.

## 1. Introduction

Periodontal tissue regeneration requires an integrated approach to the restoration of the periodontal ligament (PDL), cementum, and alveolar bone surrounding the teeth. Stem cell-based tissue engineering has made significant advances in this direction. However, these treatment approaches have had limited success in their structural and functional regeneration [1,2].

In the last decade, cell-free therapy has been considered as an alternative to cell therapy, aimed at facilitating the recruitment of resident cell populations in damaged areas by stimulating cell mobilization and return. Unfortunately, natural endogenous regenerative processes are generally limited and cannot successfully regenerate many tissues. Modern endogenous regenerative medicine uses the addition of biological agents (growth factors, exosomes) and/or biomatrices to facilitate the recruitment of resident cell populations into damaged tissues and stimulate their proliferation and differentiation. Many studies have concerned themselves with establishing biomaterials to favorably promote periodontal regeneration, in addition to surgery techniques [3,4].

Biomatrices play a critical role in the restoration of the stem cell microenvironment by providing key matrix signals for adhesion, expansion, and maintenance of self-renewal and appropriate cell differentiation [5,6]. Scaffolds of natural origin can be considered an ideal microenvironment for regeneration in terms of composition, topography, and rigidity, and they are better recognized by stem cells and more effectively stimulate their differentiation in the target tissue. ECM-derived scaffolds and ECM-mimicking scaffolds have shown promising results in facilitating constructive remodeling of various tissues in preclinical studies [7,8].

The ECM is a complex three-dimensional network of dynamically changing interconnected macromolecules that form a microenvironment that maintains cellular homeostasis during tissue formation and repair. ECM macromolecules provide structural support to cells, regulate biomechanical signaling, and ultimately provide a functional platform that determines the cellular phenotype [9]. The ECM not only maintains the structural integrity of tissues and organs but also serves as a reservoir for biochemical and biophysical signals that support cell survival, organization, and differentiation [10,11].

Proteoglycans, glycosaminoglycans, collagens, laminins (Lam), fibronectin (Fn), and some other glycoproteins are among the major components of the ECM. At the same time, each type of tissue, when examined in detail, has its own specific composition of ECM, which is secreted by the corresponding cells in the tissue and, in turn, affects the behavior of cells within the tissue. The variability in the composition of the ECM surrounding cells affects both the topology of cell–cell and cell–matrix contacts on the cell surface and the distribution of signaling biomolecules [12].

Numerous ECM analogs have been developed that mimic the physiological 3D microenvironment to support cellular function, including synthetic scaffolds derived from polymeric substrates and natural biopolymers [13,14]. Although these ECM analogs have been extensively studied for applications in tissue engineering, they lack the complex biochemical properties and three-dimensional structure of the native ECM.

During the past decade, significant progress has been made in the development of decellularization methods that allow the preservation of the native structure and composition of the ECM [10]. The main product of decellularization is the ECM, which is devoid of resident cells and retains the spatial architecture of the original tissue. The decellularized ECM (dECM) not only serves as a physical scaffold into which cells can integrate but is also able to regulate many cellular processes including migration, growth, differentiation, homeostasis, and morphogenesis. All this makes it possible to consider the dECM as a promising full-fledged biomaterial for creating tissue engineering structures to support, replace, or restore damaged tissues [15,16]. Bone dECM, for example, acts as a reservoir of pro-inflammatory cytokines, growth factors of the TGF-β family, as well as several BMPs and angiogenic growth factors such as VEGF, providing osteoinduction by regulating various phases of bone regeneration [17,18,19].

Previous studies have shown that scaffold materials themselves can trigger osteogenic differentiation of mesenchymal stem cells, demonstrating the osteoinductive potential of scaffolds [20,21]. However, there is still the problem of preserving the original composition of the ECM, especially its minor components, and evaluating its functionality for a specific tissue, with a focus on structural proteins such as collagen, Fn, Lam, glycosaminoglycans, and growth factors [22,23].

Cultivation of cells on scaffolds under 2D conditions makes it possible to study the relationship between cell functions and some components of the microenvironment. However, such a two-dimensional environment does not allow obtaining true information. It likely initiates cell behavior on the dECM that is different from the interactions that occur in native 3D tissue. Indeed, in two-dimensional cultures, many cell types show phenotypes and genotypes that are different from what occurs in vivo [24].

Compared to 2D cell cultures, cells cultured with scaffolds under 3D conditions are characterized by different interactions with ECM components and other cells; these interactions affect cell organization and cellular regulatory pathways [25]. The change in the behavior of cells in 3D cultures compared to 2D monolayers is facilitated by the structure of the ECM and the factors contained in it. Furthermore, cell suspensions in 3D hydrogels composed of ECM proteins mimic the cellular microenvironment present in vivo [26,27]. In 3D cell cultures, spatiotemporal distributions of oxygen and carbon dioxide, nutrients, and waste products are formed that regulate cell activity by simulating in vivo conditions [28]. In previous work, we have shown that the combination of decellularized tooth matrix (dTM) and decellularized PDL (dPDL) induces spontaneous differentiation of periodontal ligament stem cells (PDLSCs) in the osteogenic and odontogenic directions when the cells were cultured under 3D conditions in collagen I hydrogel [29].

As already mentioned, the tissue-specific properties of the ECM are manifested in a wide variety of matrix macromolecules. Despite the diversity of these molecules, the main constituents of the ECM are hydrated glycosaminoglycan (hyaluronic acid, HA) and fiber-forming proteins such as collagens, elastin, Fn, and Lam [30]. The exogenous addition of key ECM components (HA, Fn, and Lam) has been shown to actively influence cell adhesion, migration, proliferation, and differentiation [31,32,33,34].

The purpose of our study was to evaluate the effect of exogenous components of the ECM (HA, Fn, and Lam) on the differentiation of PDLSCs cultured with dECM (combinations of dTM and dPDL) in a 3D collagen I hydrogel. The results showed that PDLSCs cultured under these conditions exhibit phenotypic characteristics consistent with osteoblast-like and odontoblast-like cells.

## 2. Materials and Methods

### 2.1. Preparation of PDL Fragments/Strips and Tooth Crumbs/Particles

Sample collection was performed after obtaining written informed consent. A total of 1–3 molars that were free from caries and restorations were randomly collected from patients aged from 18 to 25 years under the approved guidelines set by the Ethics Committee of the A.I. Evdokimov Moscow State University of Medicine and Dentistry (protocol code 03/20 from 18 March 2020). The tooth particles and the PDL strips were prepared according to the procedures previously described [29]. Briefly, the tissues of the PDL were separated from the surface of the middle third of the tooth root under aseptic conditions. Strips 0.5–0.7 mm thick were cut and stored at −20 °C. The teeth were then washed with chlorhexidine. The teeth were sectioned at the cementum–enamel junction and particles (from 1 to 2 mm) were formed from the remaining tooth roots using an electric mill (Bosch MKM 6000, Görlingen, Germany). Quality control of samples was carried out through a dental microscope (Seiler, St. Louis, MO, USA) at a magnification of ×8.

The scheme of the experiment is shown in Figure 1.

### 2.2. Isolation of PDLSCs

PDLSCs were isolated from PDL, as previously described [29]. Briefly, the PDL strips were incubated for 70 min at 37 °C in a solution containing 2 mg/mL dispase (Gibco, Grand Island, NY, USA) and 2 mg/mL type I collagenase (Gibco, USA). The cell suspension was seeded in 6-well cell culture plates (Greiner Bio-One GmbH, Frickenhausen, Germany). Cells were grown in DMEM-GlutaMAX growth medium (Gibco, USA) and supplemented with 15% fetal bovine serum (FBS, Gibco, USA), 100 U/mL penicillin and 100 μg/mL streptomycin (Gibco, USA), and 2 mM essential amino acids (Gibco, USA) at 37 °C and 5% CO_2_. Previously characterized PDLSCs at passages 3–5 were used in experiments [29].

### 2.3. Decellularization of PDL Strips and Tooth Particles

The decellularization of the PDL strips and tooth particles was carried out by sequential incubation with a 1% SDS solution (Sigma-Aldrich, St. Louis, MO, USA), a 1% Triton X-100 solution (Sigma-Aldrich, USA), and a DNase solution (20 μg/mL; Sigma-Aldrich, USA) in 4.2 mM MgCl_2_ (Sigma-Aldrich, USA) [35]. Further, the samples were incubated in DMEM culture medium (Gibco, Thermo Fisher Scientific, USA) with antibiotics (300 U/mL penicillin, 300 μg/mL streptomycin, and 75 μg/mL amphotericin B) (Gibco, USA) and stored at −70 °C. All treatment steps were applied to samples at room temperature with constant gentle agitation of the samples in an orbital shaker (Corning™ LSE™, USA). Histological sections were stained with VECTASHIELD antifade mounting medium with DAPI (Vector Laboratories, Inc., Burlingame, CA, USA) to validate the efficiency of decellularization.

### 2.4. Preparation of Collagen I Hydrogel and Bioengineered Constructs

The collagen type I stock solution was prepared according to the procedures previously described [29].

The combination of decellularized matrices dTM and dPDL (3:1) was added to the working collagen type I solution so that the total gel volume increased by less than 25%. Then, a suspension of PDLSCs in a small volume of medium (∼50–100 µL) was added to the working collagen type I solution such that the hydrogel would contain 0.5 × 10^6^ cells/mL.

Neutralized collagen I hydrogel with the combination of dTM and dPDL embedded, the suspension of PDLSCs, with or without the ECM component (sodium hyaluronate (HA; in the final concentration 1 mg/mL, Dongkook Pharmaceutical Co., Ltd., Republic of Korea), fibronectin (Fn; 10 μg/mL, Gibco™ 33016015), laminin (Lam; 1 μg/mL, Gibco™ 23017015)) were placed in the wells of a 12-well plate in a volume of 1.5 mL per well and left to polymerize at 37 °C, 5% CO_2_ for 60 min. The most effective concentration of exogenic components of the ECM was found in preliminary studies. Neutralized collagen I hydrogel with embedded suspension of PDLSCs without the combination of dTM and dPDL with or without the ECM component served as a control.

Thus, the PDLSCs were exposed to 8 different conditions as follows: (1) Collagen I hydrogel only (Coll I); (2) Coll I + HA; (3) Coll I + Fn; (4) Coll I + Lam; (5) Coll I + dECM; (6) Coll I + dECM + HA; (7) Coll I + dECM + Fn; (8) Coll I + dECM + Lam.

After collagen polymerization, prewarmed complete DMEM-GlutaMAX (Gibco) with 10% FBS was added to the bioengineered constructs. Cultivation continued for 14 days in a CO_2_ incubator at 37 °C.

### 2.5. Histological Analysis

The 3D culture was fixed after 14 days of cultivation with 10% neutral formalin, after which it was decalcified and embedded in paraffin. Subsequently, histological sections of 4 μm were prepared, which were stained with hematoxylin-eosin according to the standard method.

### 2.6. Immunohistochemical Study

Deparaffinized sections were incubated with antibodies to CD44 (ab157107, Abcam, Cambridge, UK); STRO-1 (ab57834, Abcam), osteocalcin (OC, ab198228, Abcam), osteopontin (OPN, ab218237, Abcam), dentin sialophosphoprotein (DSPP, ab216892, Abcam), alkaline phosphatase (ALP, ab216892, Abcam), pan-cytokeratin (pan-CK, CF190321, TrueMAB™ Thermo Fisher Scientific), vimentin (Vim, MA5-11883, Thermo Fisher Scientific). Antigen demasking was performed before immunohistochemical staining using low pH EnVison FLEX Target Retrieval Solution (Dako, Glostrup, Denmark A/S) at 97 °C for 20 min. Endogenous peroxidase and host IgG were blocked. Primary antibodies were diluted according to the manufacturer’s recommendations. Incubation was carried out for 12 h at 4 °C. An EnVision FLEX detection system (Dako, Glostrup, Denmark A/S) with 2,3-diaminobenzidine DAB chromogen (DAB Chromogen Solution, Dako) was used for imaging. The nuclei were counterstained with hematoxylin. We used incubation without primary antibodies as a negative control.

### 2.7. Quantitative and Semi-Quantitative Scoring of the Immunohistochemistry Study

The expression of markers was assessed by both quantitative and semi-quantitative methods.

#### 2.7.1. Evaluation of the Expression of Various Markers in PDLSCs

Positive stain cells were counted in five images of random fields with a microscope lens magnification of 20× in ImageJ1.48 software (Wayne Rasband, National Institute of Mental Health, Bethesda, MD, USA). The number of cells in selected fields ranged from 100 to 200. The percentage of cells expressing stemness-related markers and osteogenic and odontogenic differentiation markers (*M*%) was calculated using the formula where the number of positively stained cells (*N_P_*) was divided by the total number of cells (*N_t_*):$$M\%=\frac{N_{P}}{N_{t}}\times 100\%.$$

#### 2.7.2. Semi-Quantitative Scoring of the Immunohistochemistry Study

The immunoreactivity score was determined by the intensity and distribution of the specific stain and expressed as ‘-’—no stain, ‘+-’—weak positive, ‘+’—mild or moderate focal positive, ‘++’—moderate diffuse positive, ‘+++’—strong diffuse positive. Three investigators independently evaluated all immunohistochemical preparations.

### 2.8. Statistical Analysis

Statistical significance was calculated using a one-way ANOVA with a Tukey–Kramer post hoc test for multiple comparisons (MedCalc^®^ Statistical Software version 22.009, Ostend, Belgium). Data are presented as group mean (M) ± standard error of the mean (SEM). A *p* value of less than <0.05 was considered statistically significant for all comparisons.

## 3. Results

We used a combination of dECMs (dTM and dPDL) as scaffolds to evaluate the differentiation potential of PDLSCs. The effect of individual components of the ECM (HA, Fn, and Lam) on the differentiation of PDLSCs was studied in collagen I hydrogel with dECM. PDLSCs cultured in collagen I hydrogel without the dECM with or without the addition of ECM components served as controls.

### 3.1. Morphological Characteristics

On day 2, at the beginning of cultivation, the collagen I hydrogel in bioengineered constructs contained dTM and dPDL and showed a more pronounced contraction than in the control (Figure 2A,B). However, at the end of the cultivation, on day 14, the collagen I hydrogel contraction was more pronounced in the control (Figure 2E,F).

The contraction of the collagen I hydrogel caused the formation of internal flexures filled with clusters of cells in both groups. The pattern of cell distribution in these groups was predominantly chaotic; for instance, in the subgroup with the addition of HA and in the absence of dTM and dPDL (Figure 3A), with the exception of two subgroups, where the administration of Fn and Lam in the absence of dTM and dPDL initiated a linear pattern of distribution, from the periphery to the center of the folds that formed as a result of contraction (Figure 3B,C).

### 3.2. Immunohistochemical (Phenotypic) Characterization

#### 3.2.1. Differentiation Potential of PDLSCs Cultured without dECM in Collagen I Hydrogel (Control)

Cultivation of PDLSCs in the collagen I hydrogel without the addition of ECM components revealed the expression of both stem cell markers (CD44, STRO-1) (Table 1; Figure 4A,C) and osteogenic differentiation markers such as OC, OPN (Table 1; Figure 5A,C), and ALP (Table 1; Figure 6A–D). Moreover, the expression of Vim was detected in all control groups (Table 1; Appendix A). The expression of the marker of odontogenic differentiation DSPP was not detected in all combinations without the dECM (Table 1; Figure 7A–D,I; Appendix A). Statistical analysis found differences between groups in staining for CD44, STRO-1, OC, OPN, and DSPP; data on the difference between the values in the groups are presented in Appendix A.

The addition of HA to the collagen I hydrogel increased the expression of stemness-related markers (Table 1; Figure 4A,C; Appendix A) and osteogenic differentiation markers (Table 1; Figure 5A,C; Appendix A). At the same time, in this subgroup, compared to PDLSCs cultured without the addition of ECM components, where pan-CK staining was not observed (Figure 8A), weak expression of pan-CK was found in cells located on the periphery of the collagen I hydrogel (Figure 8B).

Administration of Fn and Lam also increased the expression of osteogenic differentiation markers in PDLSCs (Table 1; Figure 5A,C; Appendix A). At the same time, if the addition of Fn only reduced the expression of stem cell markers, then the addition of Lam led to its complete absence (Table 1; Figure 4A,C; Appendix A).

#### 3.2.2. Differentiation Potential of PDLSCs Cultured with dECM in Collagen I Hydrogel

PDLSCs cultured in collagen I hydrogel with a combination of dTM and dPDL without the exogen addition of ECM components showed an increase in stemness-related (Table 1; Figure 4B,C; Appendix A) and osteogenic differentiation markers (Table 1; Figure 5B,C; Appendix A). With that, along with osteogenic differentiation, the appearance of a marker of odontogenic differentiation DSPP was noted (Table 1; Figure 7E–I; Appendix A). In addition, weak expression of pan-CK was observed in cell clusters within the flexures of the collagen I hydrogel (Figure 8E).

HA administration also enhanced the expression of stemness-related (Table 1; Figure 4B,C; Appendix A) and osteogenic differentiation markers (Table 1; Figure 5B,C; Appendix A); odontogenic differentiation was more pronounced compared to the Coll I + dECM subgroup (Table 1; Figure 7F; Appendix A). Cytokeratin-positive cells were mainly located on the surface or in the folds of the collagen I hydrogel (Figure 8F).

The addition of Fn to the collagen I hydrogel in combination with dTM and dPDL further enhanced osteogenic and odontogenic differentiation (Table 1; Figure 5B,C and Figure 7G; Appendix A). Despite the increased expression of stemness-related markers (Table 1; Figure 4B,C; Appendix A), pan-CK expression was absent (Figure 8G).

Lam also increased the expression of osteogenic differentiation markers (Table 1; Figure 5B,C; Appendix A), but the expression of odontogenic markers was less pronounced than with the addition of Fn (Table 1; Figure 7H,I; Appendix A). At the same time, as in the control group, the addition of Lam led to the disappearance of stem cell markers, despite the presence of the dECM combination (Table 1; Figure 4B,C).

Under all conditions of PDLSC cultivation in collagen I hydrogel, Vim expression was detected (Table 1; Appendix A) and also ALP expression, which increased in the presence of the dECM (Table 1; Figure 6E–H). In addition, the formation of mineralized nodules, damaged as a result of decalcification during histological processing, was observed (Figure 6C,D,F,G, arrows).

Thus, our study showed that the addition of HA, Fn, and Lam to the dECM enhanced the osteogenic and odontogenic differentiation of PDLSCs. No differences in the effect of HA and Lam on the expression of osteogenic markers were found, while the administration of Fn significantly increased the expression of osteogenic and odontogenic markers compared to HA and Lam. Lam significantly improved the odontogenic differentiation of PDLSCs compared to HA.

## 4. Discussion

Current strategies in endogenous regenerative medicine use biomatrices, the dECM in particular, to facilitate the recruitment of resident cell populations into damaged tissues and stimulate their proliferation and differentiation [36]. The dECMs containing various components of the ECM and matrix-associated factors form a microenvironment that promotes the homing of resident cells to the damaged area [8]. The dECM scaffold was approved recently by the US FDA for human therapeutic applications, making ECM scaffolds a promising therapy for human tooth regeneration [9,37]. The most important aspects of dECM are both biochemical and biomechanical properties. Current decellularization methods are still far from achieving a fully viable structure, but the production process for dECM-based scaffolds can be modified by adding growth factors and bioactive molecules. Combinations of different techniques can increase the effectiveness of decellularization and reduce the negative effects caused by the use of one technique [22]. Studies have shown that dental pulp stem cells (DPSCs) can be controlled not only by stimulation of growth factors but also through simultaneous interaction with the extracellular environment, including ECM glycoproteins [38].

In this study, we evaluated the effect of adding individual ECM components (HA, Fn, and Lam) to the dECM (a combination of dTM and dPDL) in collagen hydrogel. Cultivation of PDLSCs in collagen I hydrogel mimics in vivo conditions and induces different cell behavior when interacting with the dECM, which is absent in 2D culture [25,29].

The culturing of PDLSCs in collagen I hydrogel caused a pronounced expression of ALP, which was detected in almost all cells, regardless of the presence or absence of the dECM and any components of the ECM. Type I collagen is known to activate ALP and OPN, influencing osteoblast differentiation [21,39,40], which was confirmed in our study. In addition, the mechanical induction of cell differentiation is well known. Several in vitro studies have shown that mechanical stimulation, such as that induced by collagen I hydrogel contraction, induces differentiation of mesenchymal stem/progenitor cells into osteoblasts [41,42], which also explains the high activity of ALP. At the same time, odontoblast differentiation under the influence of mechanical forces is shown in 3D scaffolds only for DPSCs [43,44,45].

HA is a necessary component of the ECM that creates an adequate microenvironment, accelerates cell proliferation, and improves tissue healing [46]. High-molecular-weight HA is widely used in the restoration of both hard and soft tissues, especially in the form of a hydrogel, significantly affecting cell proliferation and migration, including angiogenesis. In this study, the addition of HA predictably stimulated the expression of stemness markers (CD44, STRO-1), since HA is one of the components that form the stem cell niche [47]. The appearance of the co-expression of vimentin and cytokeratin is obviously associated with this. Weak co-expression of vimentin and cytokeratin was also detected in cell clusters in the area of collagen I hydrogel folds during cultivation of PDLSCs with dECM (dTM and dPDL) without the addition of ECM components. Obviously, the use of dTM and dPDL also contributes to the maintenance of cell stemness, as evidenced by the increased expression of CD44 and STRO-1. The exogenous addition of HA did not change the expression of the osteogenic differentiation markers of PDLSCs but slightly improved their odontogenic differentiation, compared to the use of dTM and dPDL without exogenous components of the ECM. It is known that HA is often used as a tool to improve the microenvironment that promotes osteogenesis induced by BMP and TGF-β [48,49]. In this regard, we can assume that the combination of dTM and dPDL is self-sufficient for the formation of a microenvironment initiating spontaneous osteogenic differentiation.

One of the key ECM glycoproteins is Fn, which provides various cellular functions, including structural support, cell contractility, cell migration, and differentiation; in particular, it stimulates odontogenic differentiation [33]. Fn is widely used in restorative dentistry and periodontology [33]. In particular, Fn is seen as an important mediator of dentinogenesis, but only for DPSC [32,50]. Our study showed that the addition of Fn to the dECM in collagen I hydrogel without growth factors or special growth media enhanced both the osteogenic and odontogenic differentiation of PDLSCs. It is obvious that the presence of matrix-associated growth factors in the combination of dTM and dPDL provides an increase in the odontogenic differentiation of PDLSCs with the addition of Fn.

Lam is a structural and biologically active component of the ECM that promotes dentin formation and regulates odontoblast differentiation in tooth development [51,52]. Most studies are looking at the use of Lam in pulp tissue regeneration [32,53]. In our study, the exogenous addition of Lam to the combination of dTM and dPDL enhanced the expression of osteogenic differentiation markers, but the expression of odontogenic markers was less pronounced than with the addition of Fn. At the same time, Lam inhibited the expression of stemness markers (CD44, STRO-1) in collagen I hydrogel with and without the combination of dTM and dPDL. The exogenous addition of Lam is known to promote the terminal differentiation of stem cells [54,55]. Based on this, it can be assumed that the addition of Lam to the collagen I hydrogel also promotes terminal differentiation of PDLSCs.

The data obtained add to the growing body of evidence that ECM components may be an important constituent of the functional biomaterials required for the induction of phenotypic plasticity and differentiation of resident periodontal stem cells.

## 5. Conclusions

Regeneration of periodontal tissues requires an integrated approach to restore the PDL, cementum, and alveolar bone surrounding the teeth. Modern strategies of endogenous regenerative dentistry widely use biomaterials, in particular the dECM, to facilitate the recruitment of populations of resident cells into damaged tissues and stimulate their proliferation and differentiation. The most important aspects of the dECM are the biochemical and biomechanical properties that form a microenvironment that provides key matrix signals for adhesion, expansion, support for self-renewal, and appropriate cell differentiation.

This study evaluated the effect of individual components of the ECM (HA, Fn, Lam) on the odontogenic and osteogenic differentiation of PDLSCs in collagen I hydrogel using dECM (a combination of dTM and dPDL). The different potentials of these components have been demonstrated by the expression of osteogenic and odontogenic differentiation markers (OC, OPN, ALP, DSPP). This phenomenon corresponds to the conditions of the PDL in vivo, in which cells at the interface between bone and cementum tend to differentiate into osteoblasts or cementoblasts.

The addition of Fn to the dECM most effectively induces the differentiation of PDLSCs into osteoblast-like and odontoblast-like cells under 3D culture conditions. Therefore, this bioengineered construct has a high potential for future use in periodontal tissue regeneration.

## Figures and Tables

**Figure 1 cells-12-02335-f001:**
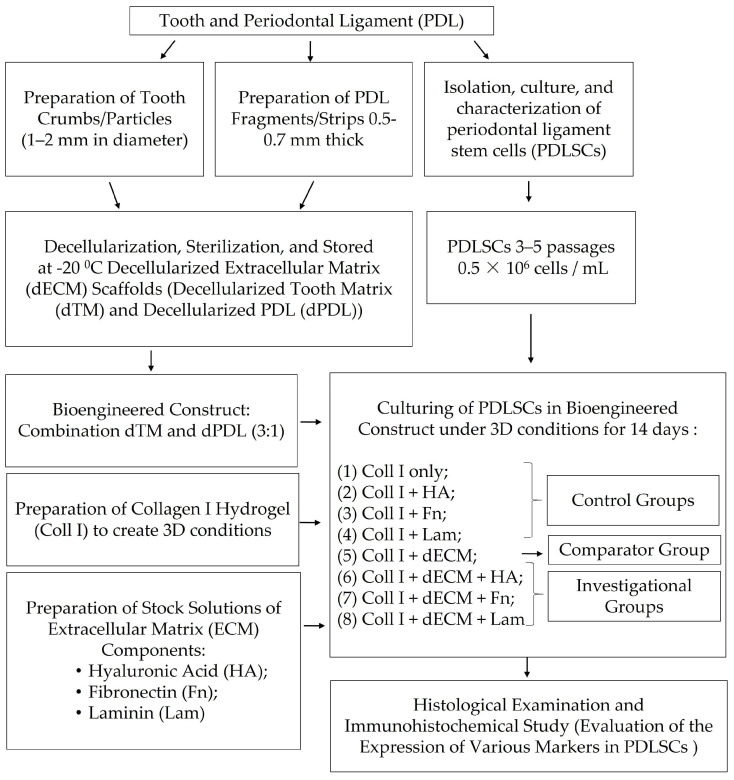
The scheme of the experiment.

**Figure 2 cells-12-02335-f002:**
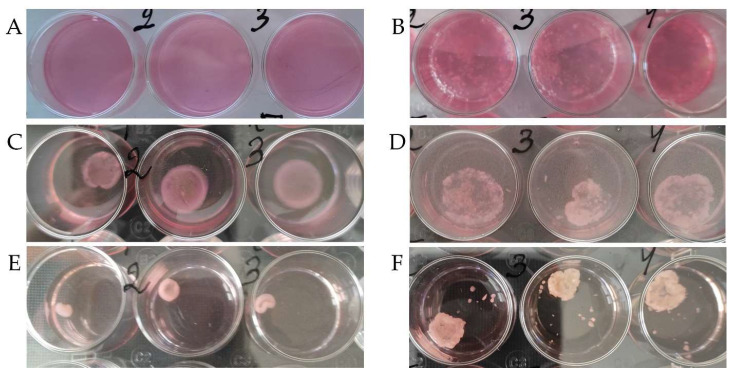
Collagen I hydrogel contraction on day 2 (**A**,**B**), day 7 (**C**,**D**), and day 14 (**E**,**F**). Three-dimensional culture: PDLSCs without scaffolds (**A**,**C**,**E**) and together with scaffolds (**B**,**D**,**F**) in a collagen I hydrogel.

**Figure 3 cells-12-02335-f003:**
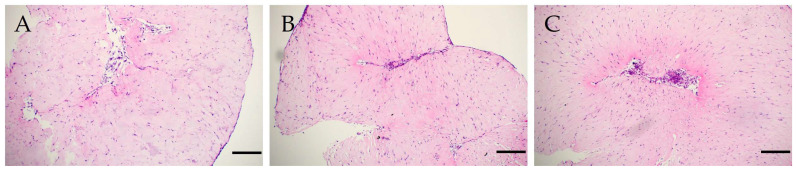
Histological sections of the bioengineered constructs in the absence of dTM and dPDL: (**A**) chaotic cell distribution in Coll I + HA; (**B**) linear pattern of cells around internal flexures of collagen I hydrogel in Coll I + Fn; (**C**) linear pattern of cells around internal flexures of collagen I hydrogel in Coll I + Lam. Hematoxylin-eosin staining. Scale bars, 200 µm in (**A**–**C**).

**Figure 4 cells-12-02335-f004:**
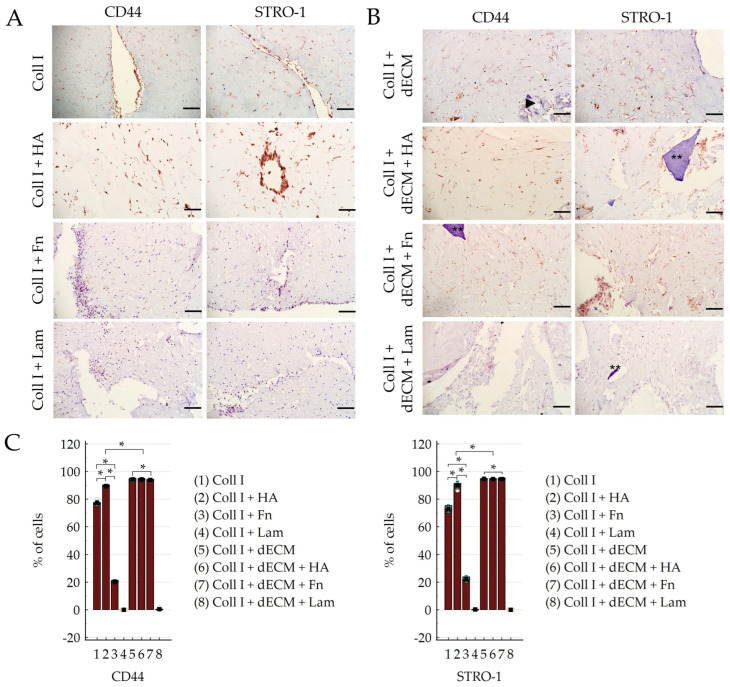
Evaluation of the expression of stem cell markers (CD44, STRO-1) in PDLSCs cultured under 3D conditions for 14 days: (**A**,**B**) Immunohistochemical staining of PDLSCs cultured without scaffolds (**A**) and with decellularized scaffolds (**B**) in collagen I hydrogel. The positive cells have a brown color. The nuclei were counterstained with hematoxylin. Two asterisks indicate dTM; an arrowhead indicates dPDL. Scale bars, 100 µm; (**C**) Morphometric analysis of the percentage of cells stained with CD44 and STRO-1 in different groups; M ± SEM, n = 5. The difference between the values in the groups was statistically significant (* *p* < 0.001).

**Figure 5 cells-12-02335-f005:**
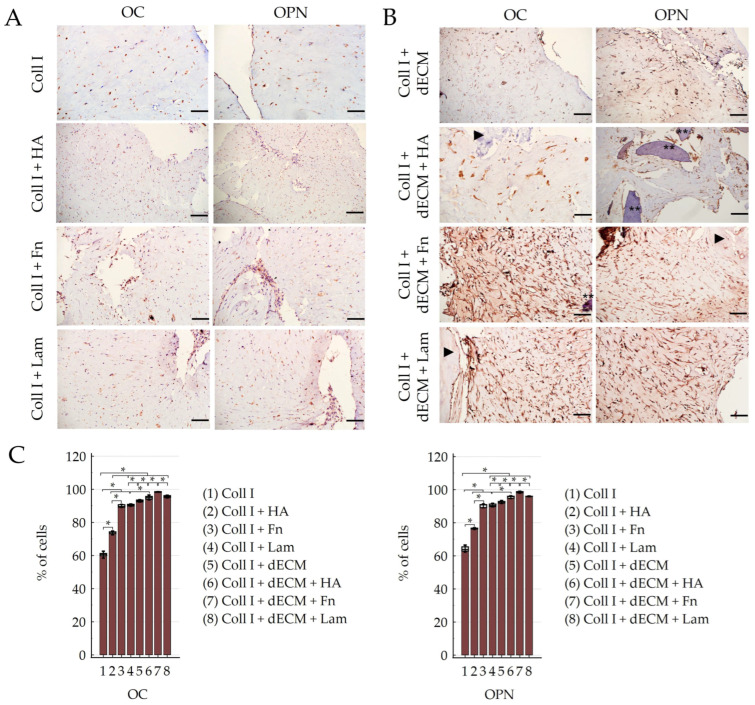
Evaluation of the expression of osteogenic (OC, OPN) differentiation markers in PDLSCs cultured under 3D conditions for 14 days: (**A**,**B**) Immunohistochemical staining of PDLSCs cultured without scaffolds (**A**) and with decellularized scaffolds (**B**) in collagen I hydrogel. The positive cells have a brown color. The nuclei were counterstained with hematoxylin. Two asterisks indicate dTM; an arrowhead indicates dPDL. Scale bars, 100 µm; (**C**) Morphometric analysis of the percentage of cells stained with OC and OPN in different groups; M ± SEM, n = 5. The difference between the values in the groups was statistically significant (* *p* < 0.001).

**Figure 6 cells-12-02335-f006:**
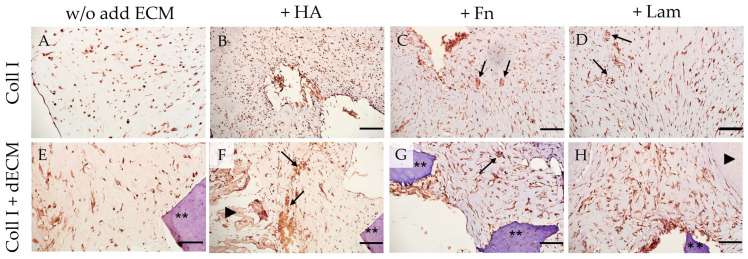
Immunohistochemical staining of PDLSCs for ALP. PDLSCs were cultured without scaffolds (**A**–**D**) and with decellularized scaffolds (**E**–**H**) in collagen I hydrogel for 14 days. The positive cells have a brown color. The nuclei were counterstained with hematoxylin. The asterisks indicate dTM; arrowheads indicate dPDL; arrows indicate the formation of mineralized nodules damaged as a result of decalcification during histological processing. Scale bars, 100 µm in (**A**,**C**–**E**,**G**,**H**), 200 µm in (**B**,**F**).

**Figure 7 cells-12-02335-f007:**
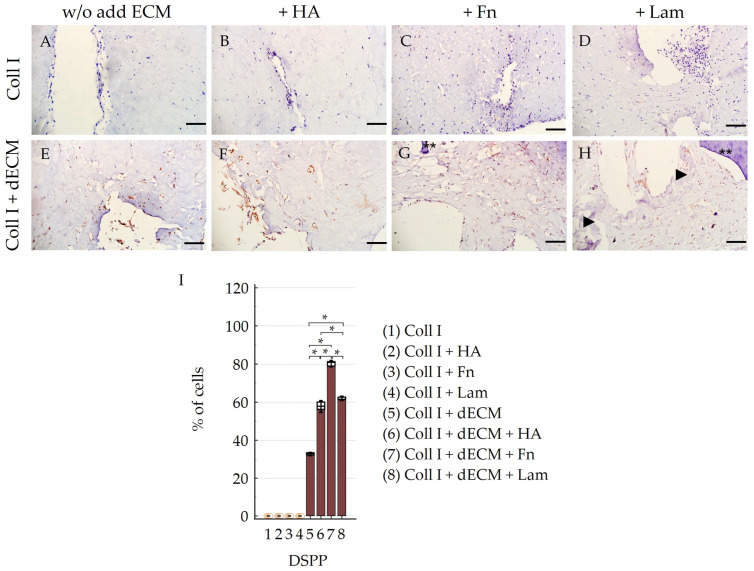
Evaluation of the expression of odontogenic (DSPP) differentiation marker in PDLSCs cultured under 3D conditions for 14 days: (**A**–**H**) Immunohistochemical staining of PDLSCs cultured without scaffolds (**A**–**D**) and with decellularized scaffolds (**E**–**H**) in collagen I hydrogel. The positive cells have a brown color. The nuclei were counterstained with hematoxylin. Two asterisks indicate dTM; arrowheads indicate dPDL. Scale bars, 100 µm; (**I**) Morphometric analysis of the percentage of cells stained with DSPP in different groups; M ± SEM, n = 5. The difference between the values in the groups was statistically significant (* *p* < 0.001).

**Figure 8 cells-12-02335-f008:**
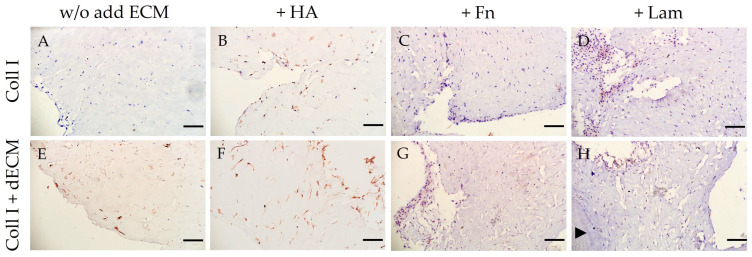
Immunohistochemical staining of PDLSCs for pan-CK. PDLSCs were cultured without scaffolds (**A**–**D**) and with decellularized scaffolds (**E**–**H**) in collagen I hydrogel for 14 days. The positive cells have a brown color. The nuclei were counterstained with hematoxylin. An arrowhead indicates dPDL. Scale bars, 100 µm. (**A**,**C**,**D**,**G**,**H**) No cell staining; (**B**,**E**) Weak staining of cells in the periphery of the collagen I hydrogel; (**F**) Moderate local expression in cells inside folds of the collagen I hydrogel.

**Table 1 cells-12-02335-t001:** Semi-quantitative scoring of the expression of various markers in PDLSCs under different culture conditions.

Condition	CD44	STRO-1	OC	OPN	DSPP	ALP	Pan-CK	Vim
Coll I	+	+	+	+	-	++	-	+++
Coll I + HA	+/++	+/++	++	++	-	++	+-	+++
Coll I + Fn	+-	+-	++	++	-	+++	-	+++
Coll I + Lam	-	-	++	++	-	+++	-	+++
Coll I + dECM	++	++	++/+++	++/+++	+	++	+-	+++
Coll I + dECM + HA	++	++	++/+++	++/+++	++	+++	+	+++
Coll I + dECM + Fn	++	++	+++	+++	+++	+++	-	+++
Coll I + dECM + Lam	-	-	+++	+++	++	+++	-	+++

‘-’—no stain, ‘+-’—weak positive, ‘+’—mild or moderate focal positive, ‘++’—moderate diffuse positive, ‘+++’—strong diffuse positive.

## Data Availability

The data presented in this study are available on request from the corresponding author. The data are not publicly available due to legal issues.

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
