# Peer review of "Influence of Extracellular Matrix Components on the Differentiation of Periodontal Ligament Stem Cells in Collagen I Hydrogel"

_cells, 2023, doi:10.3390/cells12192335_

Round 1
Reviewer 1 Report
The abstract presents a study that explores the impact of exogenous extracellular matrix (ECM) components on the differentiation of periodontal ligament stem cells (PDLSCs) cultured with decellularized ECM (dECM) within a 3D collagen hydrogel for periodontal tissue regeneration. While the study attempts to contribute to the field of endogenous regenerative dentistry, several issues arise in the manuscript.
1. Experimental design and novelty: The study appears to have included an excessive number of groups in its experimental design. This could potentially undermine the clarity and significance of the findings. Additionally, the staining results are atypical, and the differences between groups are not well-defined. The statistical analysis also lacks convincing support, with all results showing only one asterisk (*).
2. Lack of Osteogenic Differentiation Medium: One glaring concern is the absence of an osteogenic differentiation medium, typically essential for inducing osteogenic or odontoblast differentiation in stem cells. The manuscript does not mention any addition of β-glycerophosphate, ascorbic acid, or dexamethasone, which are crucial components in standard osteogenic culture media. The authors need to address how significant expression of osteogenic markers (OCN, OPN, and DSPP) was achieved without this essential induction medium, as it deviates from established practices in the field.
3. Missing representative images: The manuscript presents representative images of Colâ…¡, Colâ…¡+HA, Colâ…¡+Fn, and Colâ…¡+Lam. However, it lacks images of Colâ…¡+dECM, Colâ…¡+dECM+HA, Colâ…¡+Fn+dECM, and Colâ…¡+Lam+dECM. Including these missing images is vital to ensure a comprehensive and complete representation of the experimental findings.
4. In the material and method part, p value < 0.05 was considered to be statistically significant. While in the figure 3 legend, the difference between the values in the groups was statistically significant (*p < 0.001).
Minor editing of English language required
Author Response
We express our deep gratitude to the reviewer for the hard work of reading the paper and for all the comments and suggestions made for necessary corrections. Below is the list of all the questions and our answers to them. All changes in the manuscript are highlighted in red.
The abstract presents a study that explores the impact of exogenous extracellular matrix (ECM) components on the differentiation of periodontal ligament stem cells (PDLSCs) cultured with decellularized ECM (dECM) within a 3D collagen hydrogel for periodontal tissue regeneration. While the study attempts to contribute to the field of endogenous regenerative dentistry, several issues arise in the manuscript.
- Experimental design and novelty: The study appears to have included an excessive number of groups in its experimental design. This could potentially undermine the clarity and significance of the findings. Additionally, the staining results are atypical, and the differences between groups are not well-defined. The statistical analysis also lacks convincing support, with all results showing only one asterisk (*).
Of course, we could limit ourselves to one component of the ECM. However, we believe that a comparison of the effect of the three components of the ECM that are the most commonly used in dentistry on the differentiation of PDSCs will be of greater interest to other researchers.
The staining of cells in culture differs from the staining of cells in tissue on paraffin sections. In our case, our bioengineered construct is a tissue prototype and cell staining is absolutely typical. We believe that we have clearly shown the difference in staining in the groups.
Statistical analyses were performed using the MedCalc software package. One-way ANOVA with Tukey-Kramer post hoc test for multiple comparisons was used to determine a statistically significant difference between the previously calculated means. Statistically significant correlations between variables presented on corresponding graphs (*p < 0.001). Other information that has been shown to be statistically significant is presented in Supplementary Tables S1-S5. Due to the small size of the images, two separate stars very closely spaced can be perceived as one double star, although on all graphs between groups, the reliability is one star.
- Lack of Osteogenic Differentiation Medium: One glaring concern is the absence of an osteogenic differentiation medium, typically essential for inducing osteogenic or odontoblast differentiation in stem cells. The manuscript does not mention any addition of β-glycerophosphate, ascorbic acid, or dexamethasone, which are crucial components in standard osteogenic culture media. The authors need to address how significant expression of osteogenic markers (OCN, OPN, and DSPP) was achieved without this essential induction medium, as it deviates from established practices in the field.
Indeed, in our study, we deliberately did not add to the culture medium any components designed to induce osteogenic or odontogenic differentiation of stem cells. The role of the osteo- and odontogenic inductor in this case was played by the decellularized matrix (dECM) and individual ECM components (HA, Lam and Fn). Understanding the possible effect of dECM in various combinations with ECM components on differentiation potential was the most important goal of our study. How significant the expression of osteo- and odontogenic markers (OCN, OPN and DSPP) was achieved in the presence of the dECM and individual components (and without the use of a special osteogenic induction medium) is shown based on quantitative and semi-quantitative analysis in the "Results" section (Table 1, Figure 5, Figure 7, Supplementary Tables S3-S5).
- Missing representative images: The manuscript presents representative images of Colâ…¡, Colâ…¡+HA, Colâ…¡+Fn, and Colâ…¡+Lam. However, it lacks images of Colâ…¡+dECM, Colâ…¡+dECM+HA, Colâ…¡+Fn+dECM, and Colâ…¡+Lam+dECM. Including these missing images is vital to ensure a comprehensive and complete representation of the experimental findings.
We divided Figures 3 and 6, arranging three Figures from them, and added Figure 1. Therefore, the new Figures 4, 5 and 7 separately show stemness-related, osteogenic, and odontogenic differentiation markers, respectively, with their graphical evaluation using a diagram. The images of Coll I + dECM present in Figures 4B, 5B, 6E, 7E, and 8E; Coll I + dECM + HA – Figures 4B, 5B, 6F, 7F, and 8F; Coll I + dECM + Fn – 4B, 5B, 6G, 7G, and 8G; Coll I + dECM + Lam – 4B, 5B, 6H, 7H, and 8H.
- In the material and method part, p value < 0.05 was considered to be statistically significant. While in the figure 3 legend, the difference between the values in the groups was statistically significant (*p < 0.001).
It is corrected in Materials and Methods (page 6, line 225: «A p value of less than < 0.05 was considered statistically significant for all comparisons»).
Reviewer 2 Report
The manuscript "Influence of Extracellular Matrix Components on the Differentiation of Periodontal Ligament Stem Cells on Collagen I Hydrogel" aims to investigate the combination of ECM components with decellularized ECM in a 3D collagen hydrogel with regard to the differentiation of periodontal ligament stem cells. The manuscript is basically written in an understandable way, but it has some flaws, which I would like to elaborate in the following.
Major Concerns:
· I consider a critical point in the definition of the cell type used. The authors refer to another publication (10.3390/biom13010122), but here it is also not described in sufficient detail that only stem cells were isolated. The method described also isolates a large number of fibroblasts, the main cell type of the PDL. PDL fibroblasts can also easily differentiate into osteoblast cells. Have the cells been screened for stem cell markers or even better isolated for them (e.g., by FACS). If this is not the case, then the authors should rather speak of PDL cells (PDLC), as it is definitely a mixed culture.
· The aim of the authors is to investigate the combination of ECM components, dECM in 3D collagen gel in relation to the differentiation of PDL cells. They note: "Exogenous addition of key ECM components (HA, Fn, and Lam) has been shown to actively influence cell adhesion, migration, proliferation, and differentiation [28-31]." For the purpose of clarity, the authors should then also focus only on the corresponding HA/Ln/Lam +dECM +Coll I conditions in their presentation of results and discussion.
· Why were the morphological characteristics of the gel performed in relation to the contraction? And why were the formation of internal flexures and pattern of cell distribution observed and evaluated? This should be briefly mentioned in the results section. Furthermore, is there any statistical information available? How exactly was this evaluated? Only by visual inspection? This should be measured (size, percentage of flexures) and indicated by means of graphs.
· The authors should revise the visual presentation of their results. The evaluation of osteogenic, odontogenic and stem cell markers should be separated and considered in an isolated manner. A graphical evaluation by means of a diagram should be given for each marker and shown accordingly in the correct figure with the representative images.
· Do I understand correctly that the semi-quantitative evaluation in Table 1 represents similar experimental evaluation as Figure 3B? Then the semiquantitative evaluation should be removed. This offers no added value with regard to a quantitative cellular evaluation.
· The statistical information on the comparisons between the individual ECM components (in combination with dECM and Coll I) should be included in the main figures in corresponding graphs.
· In the discussion, the authors should focus less on repeating their findings and more on how they fit into the literature.
· The Conclusion is way too long and should be reduced to a maximum of 2-3 main statements and clinical-practical applicability.
Minor spelling issues.
Author Response
First of all, we are very grateful to the reviewer for an attentive attitude, great criticism, and certain, critical point of view regarding our paper. Undoubtedly, we take into account all the points of the review. Our answers are briefly given below. All changes in the manuscript are highlighted in red.
The manuscript "Influence of Extracellular Matrix Components on the Differentiation of Periodontal Ligament Stem Cells on Collagen I Hydrogel" aims to investigate the combination of ECM components with decellularized ECM in a 3D collagen hydrogel with regard to the differentiation of periodontal ligament stem cells. The manuscript is basically written in an understandable way, but it has some flaws, which I would like to elaborate in the following.
Major Concerns:
- I consider a critical point in the definition of the cell type used. The authors refer to another publication (10.3390/biom13010122), but here it is also not described in sufficient detail that only stem cells were isolated. The method described also isolates a large number of fibroblasts, the main cell type of the PDL. PDL fibroblasts can also easily differentiate into osteoblast cells. Have the cells been screened for stem cell markers or even better isolated for them (e.g., by FACS). If this is not the case, then the authors should rather speak of PDL cells (PDLC), as it is definitely a mixed culture.
In the present study, both PDLSC primary culture and PDLSC culture in collagen I hydrogel without the addition of ECM components (control) revealed the expression of stem cell markers CD44 and STRO-1. Expression data for both markers during cultivation in collagen I hydrogel are presented in Section 3.2.1. We agree that PDL fibroblasts can also easily differentiate into osteoblast cells. They may express the same cell surface markers as MSCs in FACS and could also differentiate into adipocytes, chondrocytes and osteoblasts (Denu R. A. et al. Fibroblasts and Mesenchymal Stromal/Stem Cells Are Phenotypically Indistinguishable. Acta Haematol 2016; 136:85–97; DOI: 10.1159/000445096). Some authors consider fibroblasts as a practical alternative to MSCs (Ichim, T.E., O’Heeron, P. & Kesari, S. Fibroblasts as a practical alternative to mesenchymal stem cells. J Transl Med 2018;16, 212. DOI:10.1186/s12967-018-1536-1). At the same time, we believe that we study PDLSCs because fibroblasts cannot express odontoblast markers and lose the surface markers of MSCs (CD44 and STRO-1) in a matrix with laminin.
- The aim of the authors is to investigate the combination of ECM components, dECM in 3D collagen gel in relation to the differentiation of PDL cells. They note: "Exogenous addition of key ECM components (HA, Fn, and Lam) has been shown to actively influence cell adhesion, migration, proliferation, and differentiation [28-31]." For the purpose of clarity, the authors should then also focus only on the corresponding HA/Ln/Lam +dECM +Coll I conditions in their presentation of results and discussion.
We believe that the results of the experimental group should be compared with those of the control group. Therefore, we discuss the results in both groups (without and with dECM).
- Why were the morphological characteristics of the gel performed in relation to the contraction? And why were the formation of internal flexures and pattern of cell distribution observed and evaluated? This should be briefly mentioned in the results section. Furthermore, is there any statistical information available? How exactly was this evaluated? Only by visual inspection? This should be measured (size, percentage of flexures) and indicated by means of graphs.
The formation of internal folds was noted only as a fact describing the microscopic picture as a result of contraction. The pattern of cell distribution is described to show that not all added components of the extracellular matrix can influence the pattern of cell distribution. We did not evaluate these data in detail morphometrically and did not perform statistical analysis, since the main aim of our work was to evaluate the effect of added matrix components on stem cell differentiation. In our opinion, a detailed morphometric assessment of the internal folds of collagen hydrogel and the nature of the distribution of cells in it would be redundant, would significantly complicate the perception of the article and distract from the main goal of the study.
- The authors should revise the visual presentation of their results. The evaluation of osteogenic, odontogenic and stem cell markers should be separated and considered in an isolated manner. A graphical evaluation by means of a diagram should be given for each marker and shown accordingly in the correct figure with the representative images.
Agreeing with the reviewer's opinion, we divided figures 3 and 6, arranging three figures from them, and added Figure 1. Therefore, the new Figures 4, 5 and 7 separately show stemness-related, osteogenic, and odontogenic differentiation markers, respectively, with their graphical evaluation using a diagram. We hope that this will improve the quality of the manuscript.
- Do I understand correctly that the semi-quantitative evaluation in Table 1 represents similar experimental evaluation as Figure 3B? Then the semiquantitative evaluation should be removed. This offers no added value with regard to a quantitative cellular evaluation.
Semi-quantitative assessment in table 1 reflects the relative immunoreactivity of cells, that is, the expressiveness of the immunohistochemical reaction. Figure 3B (Figures 4C, 5C and 7C in the updated version of the manuscript) presents a quantitative analysis of the percentage of cells staining positively against target markers, regardless of staining intensity.
- The statistical information on the comparisons between the individual ECM components (in combination with dECM and Coll I) should be included in the main figures in corresponding graphs.
We divided figures 3 and 6, arranging three figures from them, and added Figure 1. Therefore, the new Figures 4, 5 and 7 separately show stemness-related, osteogenic, and odontogenic differentiation markers, respectively, with their graphical evaluation using a diagram.
- In the discussion, the authors should focus less on repeating their findings and more on how they fit into the literature.
We added the additional references in “Discussion”.
The Conclusion is way too long and should be reduced to a maximum of 2-3 main statements and clinical-practical applicability.
We agree that the Conclusions section may not have been very concise. However, the reduction of this section to 2-3 statements does not coincide with the opinion of other reviewers, who, on the contrary, propose to expand the conclusions even more. Nevertheless, we slightly corrected some of the wording, shortening and presenting the goal and results of the study in a more lapidary style. Since our version does not distort the meaning of the "Conclusions" section and contains a summary of the work done and the prospects for the future, we reserve the right to leave this option as a compromise between the opinions of the reviewers.
Reviewer 3 Report
Authors have studied a relevant area of the effect of exogenous components of ECM, particularly HA, laminin, and Fibronectin, on the differentiation of PDL stem cells cultured with dECM in a collagen hydrogel. The study is well conceptualized and executed. The findings have been presented well.
Author Response
We express our deep gratitude to the reviewer for his work of reading the paper.
Reviewer 4 Report
The article entitled “Influence of Extracellular Matrix Components on the Differentiation of Periodontal Ligament Stem Cells on Collagen I Hydrogel”. The aim of this study was to evaluate the effect of exogenous components of extracellular matrix (hyaluronic acid, laminin, fibronectin) on the differentiation of periodontal ligament stem cells (PDLSCs) cultured with dECM (combinations of decellularized tooth matrices and periodontal ligament) in a 3D collagen hydrogel.
Below are some suggestions:
In the Abstract:
- The abstract is well written and objective, but my suggestion is to insert the methodology, an important point, since it is an experimental study.
1. In the Introduction:
- The introduction is clear, describing the importance of the clinical importance, the biomatrices and the objective of the research.
2. Materials and Methods:
- The methodology is well described, but I suggest an experimental design at the beginning, summarizing all the steps carried out in the research. In addition, an experimental design brings images and figures, providing a visualization of the methodology.
3. Results:
- 3.1. Morphological characteristics: I suggest that the authors adjust and improve the description of the legend of Figure 1 and 2.
4. Discussion:
- The discussion is correct and well written, comparing the results found in the study with data from the literature, I only suggest adding the limitations of the research.
5. Conclusion:
- The conclusion is well written, bringing a summary of what was done in the research and future perspectives.
**I suggest authors insert a greater number of references for a better scientific basis.
Moderate editing of English.
Author Response
First of all, we are very grateful to the reviewer for the attentive attitude to our paper. Thank you for the specified shortcomings, recommendations for changes, additions, and corrections. All changes in the manuscript are highlighted in red.
The article entitled “Influence of Extracellular Matrix Components on the Differentiation of Periodontal Ligament Stem Cells on Collagen I Hydrogel”. The aim of this study was to evaluate the effect of exogenous components of extracellular matrix (hyaluronic acid, laminin, fibronectin) on the differentiation of periodontal ligament stem cells (PDLSCs) cultured with dECM (combinations of decellularized tooth matrices and periodontal ligament) in a 3D collagen hydrogel.
Below are some suggestions:
In the Abstract:
- The abstract is well written and objective, but my suggestion is to insert the methodology, an important point, since it is an experimental study.
We took this comment into account and made changes. Regarding the justification, we followed the current Author Guidelines for the Cells, which say ‘Abstract: A single paragraph of about 200 words maximum’.
- In the Introduction:
- The introduction is clear, describing the importance of the clinical importance, the biomatrices and the objective of the research.
- Materials and Methods:
- The methodology is well described, but I suggest an experimental design at the beginning, summarizing all the steps carried out in the research. In addition, an experimental design brings images and figures, providing a visualization of the methodology.
We added Figure 1 (the scheme of the experiment).
- Results:
- 3.1. Morphological characteristics: I suggest that the authors adjust and improve the description of the legend of Figure 1 and 2.
In agreement with the reviewer's opinion, we changed the description of the legend of Figures 1 and 2. Since we added Figure 1, the numbering of the figures has changed, now figures 1 and 2 have become 2 (page 6, lines 239-241) and 3 (page 7, lines 251-254), respectively.
- Discussion:
- The discussion is correct and well written, comparing the results found in the study with data from the literature, I only suggest adding the limitations of the research.
- Conclusion:
- The conclusion is well written, bringing a summary of what was done in the research and future perspectives.
**I suggest authors insert a greater number of references for a better scientific basis.
We added the additional references in “Introduction” and “Discussion”.
Reviewer 5 Report
Dear authors,
We have read with interest your manuscript investigating the effects of some ECM components. The overall project is well conducted, leading to reliable conclusions.
Here are some remarks-questions :
> Affiliation : "3 Department of Paradontology": please correct
> Materials & Methods: "refrigerator at -20" should be freezer
> Introduction: the introduction lacks clear objectives of the study. On the contrary, it contains a very short summary of the results that should be placed in the 'Conclusion'. Some objectives should appear at the end of the 'Introduction' section, helping the reader to follow the research work.
> Conclusions: the aim of the study should be moved to 'Introduction' (and short summary should be placed here.
> General question: you mention several times "bioengineered construct", and highlight the impact of your research on tissue engineering. Could your construct be considered as "organoid"? Would you consider using your bioengineered construct as organoid, grafted into periodontal defects? It would be interesting if your manuscript open toward the use of such organoid / bioengineered construct.
Yours faithfully,
Author Response
We express our deep gratitude to the reviewer for the work of reading and all the comments and suggestions made. Below – the list of questions and our answers for each of them. All changes in the manuscript are highlighted in red.
Dear authors,
We have read with interest your manuscript investigating the effects of some ECM components. The overall project is well conducted, leading to reliable conclusions.
Here are some remarks-questions :
> Affiliation : "3 Department of Paradontology": please correct
It is just a misspell, it is corrected.
> Materials & Methods: "refrigerator at -20" should be freezer
It is just a misspell in Materials and Methods, it is corrected. Minor grammatical errors and typos have been corrected through the text.
> Introduction: the introduction lacks clear objectives of the study. On the contrary, it contains a very short summary of the results that should be placed in the 'Conclusion'. Some objectives should appear at the end of the 'Introduction' section, helping the reader to follow the research work.
Agreeing with the reviewer's opinion, we changed the section “Inroduction” (page 3, lines 119-123) and “Conclusion” (page 14, lines 449-453).
> Conclusions: the aim of the study should be moved to 'Introduction' (and short summary should be placed here.
Agreeing with the reviewer's opinion, we changed the section “Inroduction” (page 3, lines 119-123) and “Conclusion” (page 14, lines 449-453). We hope that this will improve the quality of the manuscript.
> General question: you mention several times "bioengineered construct", and highlight the impact of your research on tissue engineering. Could your construct be considered as "organoid"? Would you consider using your bioengineered construct as organoid, grafted into periodontal defects? It would be interesting if your manuscript open toward the use of such organoid / bioengineered construct.
We cannot consider our bioengineered construct as “organoid” because it contains only decellularized matrixes in the collagen gel without cells. While an organoid is “a self-organized 3D tissue that is typically derived from stem cells, and which mimics the key functional, structural and biological complexity of an organ”.
Round 2
Reviewer 2 Report
The authors have responded to all my criticisms accordingly and have incorporated them to a certain extent. Of course, I understand that there are criticisms that the authors do not want to include in their manuscript (e.g. group reduction). However, I still see this as the biggest weakness of the manuscript in terms of clarity and reader-friendliness. Nevertheless, I think that the quality has improved significantly and can agree to a publication.